# Rabies transmitted from vampires to cattle: An overview

**Diego Soler-Tovar**[1,2], **Luis E. Escobar**[1,2,3,4,5]*

**1** Department of Fish and Wildlife Conservation, Virginia Tech, Blacksburg, VA, United States of America,
**2** Faculty of Agricultural Sciences, Universidad de La Salle, Bogotá, Colombia, **3** Kellogg Center for
Philosophy, Politics, and Economics, Virginia Tech, Blacksburg, VA, United States of America, **4** Global
Change Center, Virginia Tech, Blacksburg, VA, United States of America, **5** Center for Emerging Zoonotic
and Arthropod-borne Pathogens, Virginia Tech, Blacksburg, VA, United States of America

* escobar1@vt.edu

## Abstract

Rabies is a zoonotic infectious disease of global distribution that impacts human and animal
health. In rural Latin America, rabies negatively impacts food security and the economy due
to losses in livestock production. The common vampire bat, *Desmodus rotundus*, is the
main reservoir and transmitter of rabies virus (RABV) to domestic animals in Latin America.
*Desmodus rotundus* RABV is known to impact the cattle industry, from small farmers to
large corporations. We assessed the main patterns of rabies in cattle attributed to *D. rotundus* RABV across Latin America. Epidemiological data on rabies from Latin America were
collected from the Pan American Health Organization spanning the 1970–2023 period.
Analyses revealed an average of 450 outbreaks annually for the countries where *D. rotundus* is distributed, with at least 6 animals dying in each outbreak. Brazil, Colombia, Peru,
and Mexico were the Latin American countries with the highest number of rabies outbreaks
during the study period and are the most affected countries in recent years. Findings suggest a re-emergence of bat-borne rabies in the region with more outbreaks reported in
recent years, especially during the 2003–2020 period. Rabies outbreaks in cattle in the
2000–2020 period were significantly more frequent than in previous decades, with an
increase in cross-species transmission after 2002. The size of outbreaks, however, was
smaller in recent years, involving lower cattle mortality. Peru, El Salvador, and Brazil
showed a strong association (R = 0.73, *p* = 0.01) between rabies incidence in *D. rotundus*
(rates per million humans: 1.61, 0.94, and 1.09, respectively) and rabies outbreaks in cattle
(rates per million cattle: 465.85, 351.01, and 48.22, respectively). A sustained, standardized, and widespread monitoring of *D. rotundus* demography and health could serve to
inform an early warning system for the early detection of RABV and other bat-borne pathogens in Latin America. Current data can be used to forecast when, where, and in which
intensity RABV outbreaks are more likely to occur in subtropical and tropical Latin America.
A decrease in the size of outbreaks could suggest that strategies for epidemic management
(e.g., education, early diagnosis, vaccination) have been effective. The increase in the number of outbreaks could suggest that the factors facilitating cross-species transmission could
be on the rise.

transmitted from vampires to cattle: An overview.
PLoS ONE 20(1): e0317214. https://doi.org/

de Colombia, COLOMBIA

**Data Availability Statement:** The raw data of the
study are available as open access at SIRVERA:
https://sirvera.panaftosa.org.br/. The data derived
from the analysis, as well as the R code used for
the analysis, will be available once the manuscript
is accepted.

**Funding:** LEE was supported by National Science Foundation CAREER (2235295) and HEGS (2116748) awards. Research reported in this publication was supported by the National Institute of Allergy and Infectious Diseases of the National Institutes of Health under Award Number K01AI168452. The content is solely the responsibility of the authors and does not necessarily represent the official views of the National Institutes of Health. There was no additional external funding received for this study.

**Competing interests:** The authors have declared that no competing interests exist.

## Introduction

Rabies is a wildlife disease causing negative impacts on ecosystem, animal, and human health [1]. Rabies generates deaths in humans and pets, direct and indirect economic losses to livestock, especially cattle in Latin America, and localized extinctions of endangered wildlife due to rabies control (e.g., bat culling) and outbreaks (e.g., Ethiopian wolves) [1–4]. Rabies is a highly lethal, neglected tropical, zoonotic disease caused by rabies virus (RABV; *Lyssavirus*) [5]. RABV is generally transmitted through the saliva of infected mammals [6]. RABV can also be transmitted through scratching, licking, inhalation of aerosols, and, rarely, through organ transplants [1].

Rabies transmitted by the common vampire bat, *Desmodus rotundus*, in Latin America and the Caribbean presents a complex epidemiological situation that requires multidisciplinary efforts for its understanding and control [7, 8]. Bats are recognized as natural reservoirs of diverse RABV variants. While canine rabies has decreased and is under control in Latin America, bat-borne rabies is re-emerging [9].

In subtropical and tropical areas of Latin America, *D. rotundus* is the natural reservoir of RABV variant 3 [10, 11]. Other wild animals, such as coaties and non-human primates, can be infected and transmit RABV [12, 13]. Cattle are the most preferred prey of *D. rotundus* [14, 15], which generates considerable economic losses to agriculture [16, 17]. The expansion of the cattle industry increases prey availability for *D. rotundus*, altering the sylvatic RABV transmission cycle [17].

In the Neotropics, from southern Mexico to central Chile and Argentina, RABV is the most important bat-borne pathogen [18–20]. RABV genetic lineages from bats have been found in dogs and cats, demonstrating the epidemiological role of bats in the transmission of RABV to domestic animals [21]. RABV bat variants in humans has recently increased [22]. The large number of bat species, the impossibility of vaccinating them, and their roosting and feeding behavior highlight the challenge of controlling RABV in bats [21, 23].

The economic impacts of rabies in cattle include direct mortality, direct and indirect losses in dairy and meat production, and prevention and control costs [24]. For example, in a study conducted in Brazil, the estimated annual economic loss was at least USD 5000 per farm [25]. Additionally, estimated amounts spent on vaccination range between USD 2 and 1437 per property, depending on the number of livestock [25]. The average cost of annual vaccination per farm is USD 148 [25]. The relationship between the estimated cost of rabies vaccination of the entire herd and the economic losses per property could be, on average, 9.74% [25]. The relationship between the total cost of vaccination and the total economic loss, adding all properties, could be at least 5.8% [25].

*Desmodus rotundus* is one of the only three bat species that feed exclusively on blood [26]. *Desmodus rotundus* has feeding habits that vary with changes in the availability of wild and domestic prey across its range [17, 23, 27, 28]. The intensification of livestock breeding has created a new, abundant, and reliable blood source for *D. rotundus* [26, 29]. The increase in prey resources has caused *D. rotundus* population growth and range expansion [15, 30]. The geographical expansion of *D. rotundus* during the last century is linked to the increase in availability of livestock and shelters, such as mines, tunnels, wells, sewers, and abandoned houses [31]. The wild prey of *D. rotundus* has been displaced due to resource extraction and agricultural expansion in Latin America, modifying *D. rotundus'* feeding preferences [8].

The objective of this study was to explore *D. rotundus* RABV incidence in livestock in Latin America. Considering the sustained expansion of agriculture in this region [32] hindered by the continuous efforts to control *D. rotundus* colonies [23], we hypothesized that the burden of *D. rotundus* RABV in cattle has remained stable across time. Results from this study could

help determine where and when *D. rotundus* RABV prevention should be prioritized trough vaccination.

## Materials and methods

### Data

The study area included 21 continental/inland Latin American countries where *D. rotundus* is naturally distributed. Countries studied included (in latitudinal order) Mexico, Belize, Guatemala, Honduras, El Salvador, Nicaragua, Costa Rica, Panama, Colombia, Venezuela, Guyana, Suriname, French Guiana, Ecuador, Brazil, Peru, Bolivia, Chile, Argentina, Paraguay, and Uruguay. Data on *D. rotundus* RABV outbreaks in cattle were collected from the Regional Information System for the Epidemiological Surveillance of Rabies (SIRVERA) during the 1970–2023 period [33]. SIRVERA is a database for rabies prevention in the Americas where national health authorities of countries in the Americas report rabies incidence monthly in coordination with PANAFTOSA-PAHO/WHO [34]. Original SIRVERA data for this period were composed of more than 54,145 records for a plethora of species, our study focused on rabies in cattle which represented most of the records (45.2%). The inclusion criteria for the SIRVERA data included reports with information on types of cases (i.e., animals), date of notification (i.e., between January 1970 to December 2023), and target species (i.e., cattle).

### Analyses

Spatial analyses were carried out using choropleth maps to visualize area patterns [35]. Country, year, month, and number of affected cattle (cases) were evaluated, and each report in the database was considered an outbreak. In addition, monthly seasonality analyses and heat maps were made to describe the distribution of rabies cases in cattle by country accounting for their latitude [36]. Change in annual frequency of rabies outbreaks over time was estimated by country. Change of incidence were estimated by comparing annual values against a baseline from the 1970–1979 period. A value equal to 0 denoted no change, >0 an increase, and <0 a decrease annual incidence of outbreaks.

Outbreaks were defined as at least one animal reported infected, while cases were defined as the actual number of animals affected during an outbreak or period. Linear regression models were performed to quantify the relationship between the number of RABV-infected *D. rotundus* and RABV outbreaks in cattle. Rabies incidence was estimated annually for *D. rotundus* and cattle. To mitigate sampling bias, rabies reports in *D. rotundus* were standardized according to the total human population per country because there are no national population data for the bat species. Rates of rabies reports in cattle were standardized according to the total cattle population by country. Rabies in *D. rotundus* and cattle were presented per million individuals to have comparable units (S1 Table). Software R (R version 4.3.1) and R Studio (Version 2023.06.1+524) were used for the statistical analysis and data visualization. R packages employed included *dplyr*, *tidyr*, and *devtools* for data manipulation, *ggplot2*, *bbplot*, *ggalt*, *ggpubr*, and *ggrepe* for data visualization and analysis [37], and *sp*, *rgdal* and *maptools* for spatial analyses [38].

## Results

The total number of rabies outbreaks in cattle in Latin America during the study period was 23,869. Of the country records of rabies outbreaks in cattle available in SIRVERA, 8.2% (n = 1958) did not have administrative data at the state, provincial, or departmental level, and 23.9% (n = 5709) did not have data at the municipal level. The highest accumulation of

**Table 1. Number of rabies outbreaks in cattle per country (in latitudinal order) between 1970 and 2023.** Outbreaks: The total number of outbreaks for the study period by country. Cases: The total number of cases for the study period by country. Cases/Outbreak: average of cases per outbreak. Outbreaks/Year: average of outbreaks per year. Cases/Year: cases per year for each country. Bold: highest values for each variable. (Data source: SIRVERA).

| Country | Outbreaks | Cases | Cases/Outbreak | Outbreaks/Year | Cases/Year |
|---|---|---|---|---|---|
| Mexico | **2,286** | **16,161** | 7.1 | **57.2** | **304.9** |
| Belize | 182 | 231 | 1.3 | 7.6 | 4.4 |
| Guatemala | 412 | 797 | 1.9 | 8.9 | 15.1 |
| Honduras | 167 | 785 | 4.7 | 4.2 | 14.8 |
| El Salvador | 276 | 1138 | 4.1 | 6.9 | 21.5 |
| Nicaragua | 102 | 335 | 3.3 | 2.4 | 6.3 |
| Costa Rica | 143 | 202 | 1.4 | 3.7 | 3.8 |
| Panama | 233 | 992 | 4.3 | 5.8 | 18.7 |
| Colombia | **2,858** | **5,227** | 1.8 | **58.3** | **98.6** |
| Venezuela | 310 | **4,220** | **13.6** | 6.7 | **79.6** |
| Guyana | 35 | 546 | 15.6 | 2.7 | 10.3 |
| Suriname | 11 | 52 | 4.7 | 1 | 0.9 |
| French Guiana | 5 | 19 | 3.8 | 1 | 0.4 |
| Ecuador | 894 | 1600 | 1.8 | 17.9 | 30.2 |
| Brazil | **11,302** | **87,515** | 7.7 | **226.1** | **1651.2** |
| Peru | **2,731** | **4,089** | 1.5 | **56.9** | **77.2** |
| Bolivia | 566 | 2,111 | 3.7 | 14.2 | 39.8 |
| Chile | 9 | 32 | 3.6 | 1 | 0.6 |
| Argentina | 338 | **4,935** | **14.6** | 8.2 | **93.1** |
| Paraguay | 961 | 2,722 | 2.8 | 19.2 | 51.4 |
| Uruguay | 48 | 92 | 1.9 | 9.6 | 1.7 |
| Average | 1136.6 | 6371.5 | **5.1** | 24.7 | 120.2 |
| Total | 23869 | 133801 | **5.6** | 450.4 | 2524.5 |

outbreaks in the period 1970–2023 occurred in Brazil (11,302 outbreaks), followed by Colombia (2858 outbreaks), Peru (2731 outbreaks), and Mexico (2286 outbreaks). For the rest of Latin America, the accumulation of outbreaks was less than 900 total outbreaks reported per country during the entire study period (Table 1, Fig 1A).

A total of 133,801 heads of cattle were reported infected with RABV (rabies cases) in Latin America during 1970–2023. The highest number of cases occurred in Brazil (87,515) and Mexico (16,161), which together accumulated 77.5% of all rabies-driven cattle cases in Latin America during the study period (Table 1, Fig 1B). Outbreak size varied geographically, with values ranging from 1 to 5900 cases per outbreak for Brazil, 1 to 1535 cases per outbreak for Mexico, and from 1 to 1360 cases per outbreak for Argentina. There was a variable number of total number of cases per country during the study period (Table 1).

Rabies outbreaks in cattle across Latin America were stable across the year, with a sustained monthly incidence of outbreaks (Fig 2A and 2B). The average number of outbreaks across the study area in January was 111.1 (±53.2), February 115.8 (±52.8), March 109.5 (±51,9), April 108.7 (±51.2), May 123.9 (±61.1), June 117.8 (±58.7), July 105.4 (±52,1), August 115.7 (±59.4), September 104.3 (±51.2), October 101.5 (±49,4), November 88.9 (±44.7), and December 76.9 (±36.3). There was non-significant monthly tendency in the number of outbreaks to decrease late in the year with a weak peak of in May (Fig 2B). We found an overall average of 106.6 outbreaks per month during the study period across Latin America. On average, 450.4 rabies outbreaks/year were reported across Latin America in the 50 years assessed, but this incidence differed significantly (Anova: $F_{(20,707)} = 11.66$, $p < 0.001$) across countries (Table 1). The total

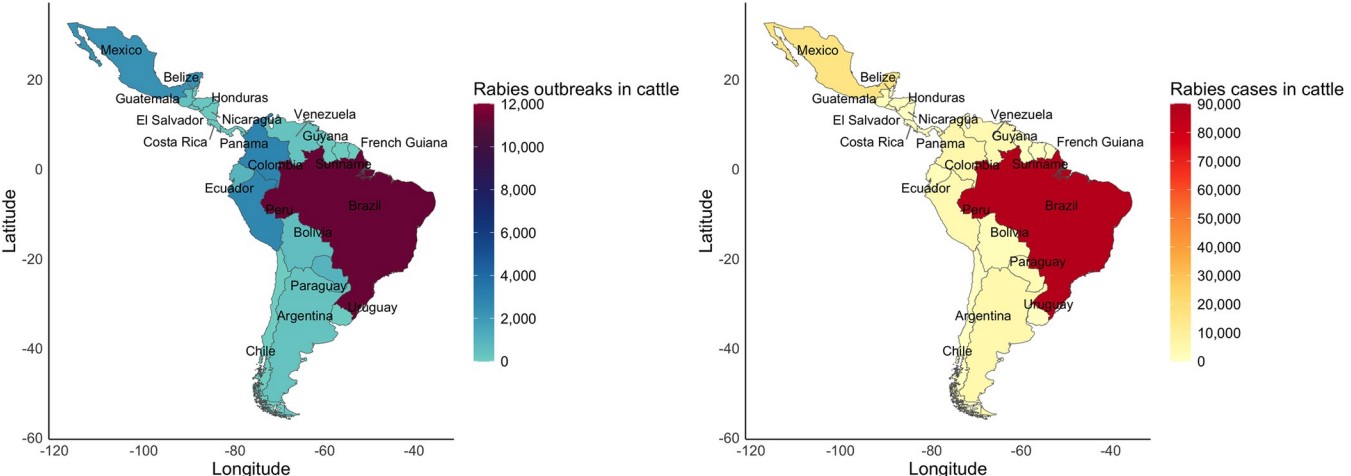

**Fig 1. Geographic distribution of rabies outbreaks in cattle between 1970 and 2023. A)** Map of the total number of outbreaks per country denoting counties with the highest (red) and lowest (blue) number of rabies outbreaks in cattle. **B)** Map of the total number of cases of rabies in cattle.

number of outbreaks for the 1970s decade was 164, while for the 2010s it reached 11,787. On the contrary, the total number of cases in the 1970s was 32,935, while for the 2010s it dropped to 15,649 cases per decade (S1 Fig).

Countries in the tropical belt, close to the Equator line (e.g., Brazil, Colombia, southern Mexico, and Peru), presented the highest numbers of outbreaks (Fig 2A). Countries with the highest numbers of outbreaks included Brazil 226.1 (±48.9) outbreaks/year, Colombia 58.3 (±11.9) outbreaks/year, Mexico 57.2 (±14.7) outbreaks/year, and Peru 56.9 (±11.1) outbreaks/year (Fig 2C). The average number of rabies cases per outbreak was 5.1 when accounting for the entire study area and period. Argentina (14.6 cases/outbreak) and Venezuela (13.6 cases/outbreak) had the highest outbreak sizes with ~14 dead animals per outbreak during the 50 years. The overall average number of outbreaks per year across the study area was 24.7. Controlling frequency of outbreaks by cattle density per country revealed some countries with high burden of rabies not detected in when using overall outbreak numbers. For example, Peru had 466 outbreak reports per million cattle and El Salvador 351 outbreak reports per million cattle, while Brazil had only 48 outbreak reports per million cattle.

The average number of cases annually was 120.2, with Brazil (1651.2), Mexico (304.9), Colombia (98.6), Argentina (93.1), and Peru (77.2) showing the highest annual incidence (Table 1). A significant positive relationship was observed between annual outbreaks and annual cases in cattle in Brazil, Mexico, Colombia, and Peru. For other countries, the annual number of cases did not correlate with the number of outbreak events (S2 Fig).

Outbreak incidence of rabies in cattle, relative to the 1970–1979 baseline period, increased per year from 1980 to 2023 in Latin America, with a heterogeneous but sustained increase in outbreaks in all countries since 2002. Brazil showed a peak of 1061 outbreaks in 2006 compared with its 1970s baseline (standardized with a value of 0), followed by Mexico with a peak (375 outbreaks) in 2005, and Colombia with a peak (265 outbreaks) in 2009. All countries showed a tendency to decrease in the annual number of outbreaks early in the 2020s decade (Fig 3).

Only 10 countries, (i.e., Peru, El Salvador, Brazil, Bolivia, Paraguay, Uruguay, Panama, Mexico, Argentina, and Colombia) of the 21 analyzed, reported rabies in *D. rotundus* (S1 Table). For the countries with *D. rotundus* incidence data, there was a positive association

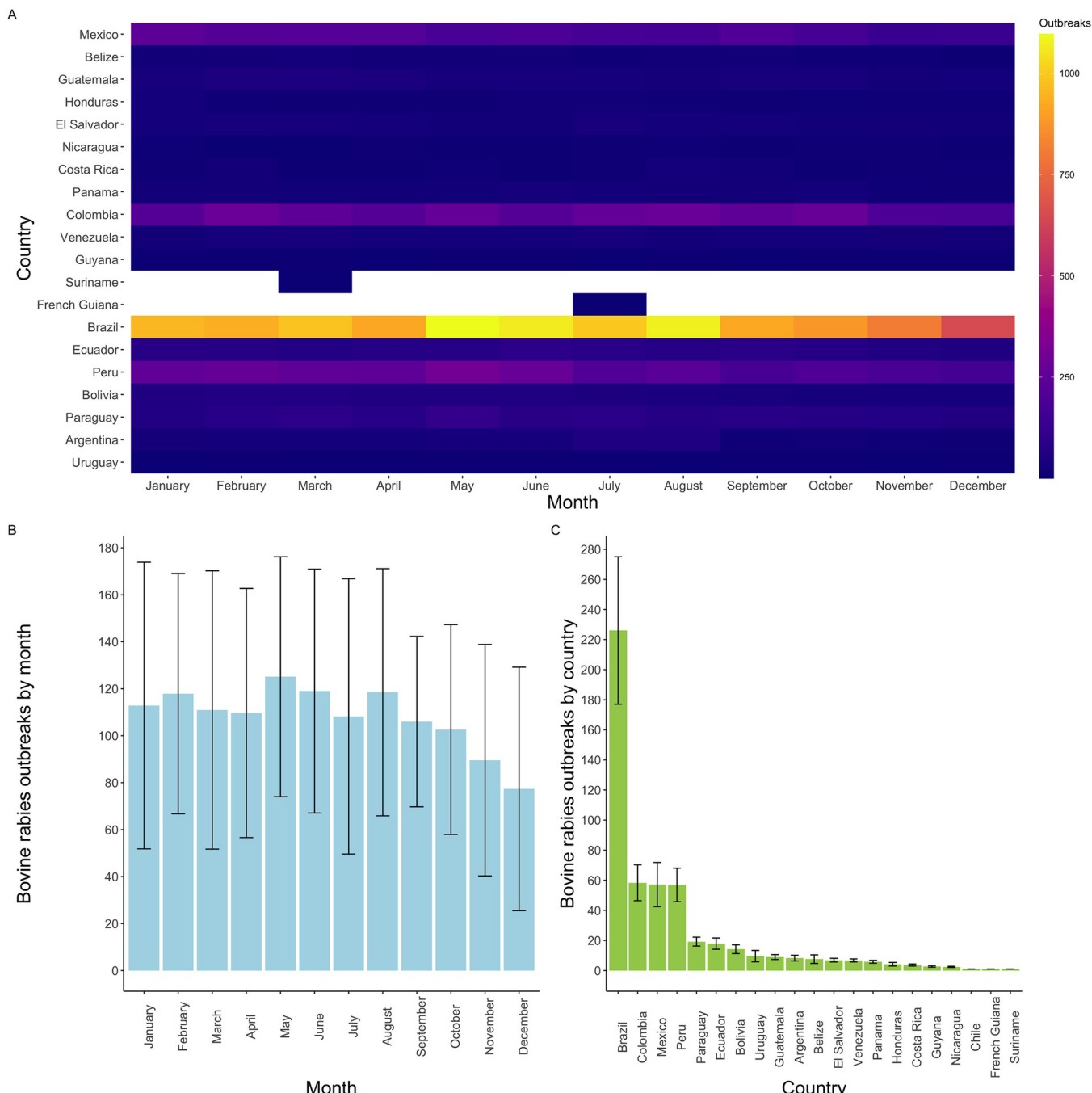

**Fig 2. Monthly and annual rate of cattle rabies occurrence in Latin America between 1970 and 2023. A)** Heatmap of total cattle rabies outbreaks reported by month (*x*-axis) and country (*y*-axis) ranked according to latitude. Note the peak in outbreaks in countries in tropical latitude (i.e., Brazil, Peru, Colombia). **B)** Monthly average of cattle rabies outbreaks (bars) and standard error of the mean (black lines). Note the lack of significant differences among months, with a tendency to decrease in variation in September and an increase in variation in the December-January period. **C)** Annual average of cattle rabies outbreaks by country (green bars) and standard error of the mean (black lines) (black lines). Brazil has the highest incidence (>220 outbreaks/year), followed by Colombia, Mexico, and Peru (~55 outbreaks/year), and Paraguay and Ecuador (~20 outbreaks/year).

between the number of rabid *D. rotundus* and the number of rabies outbreaks in cattle (R = 0.73, *p* = 0.01). Peru, El Salvador, Panama, and Colombia showed more outbreaks of rabies in cattle than expected by the number of rabid *D. rotundus* reported (Fig 4, S1 Table).

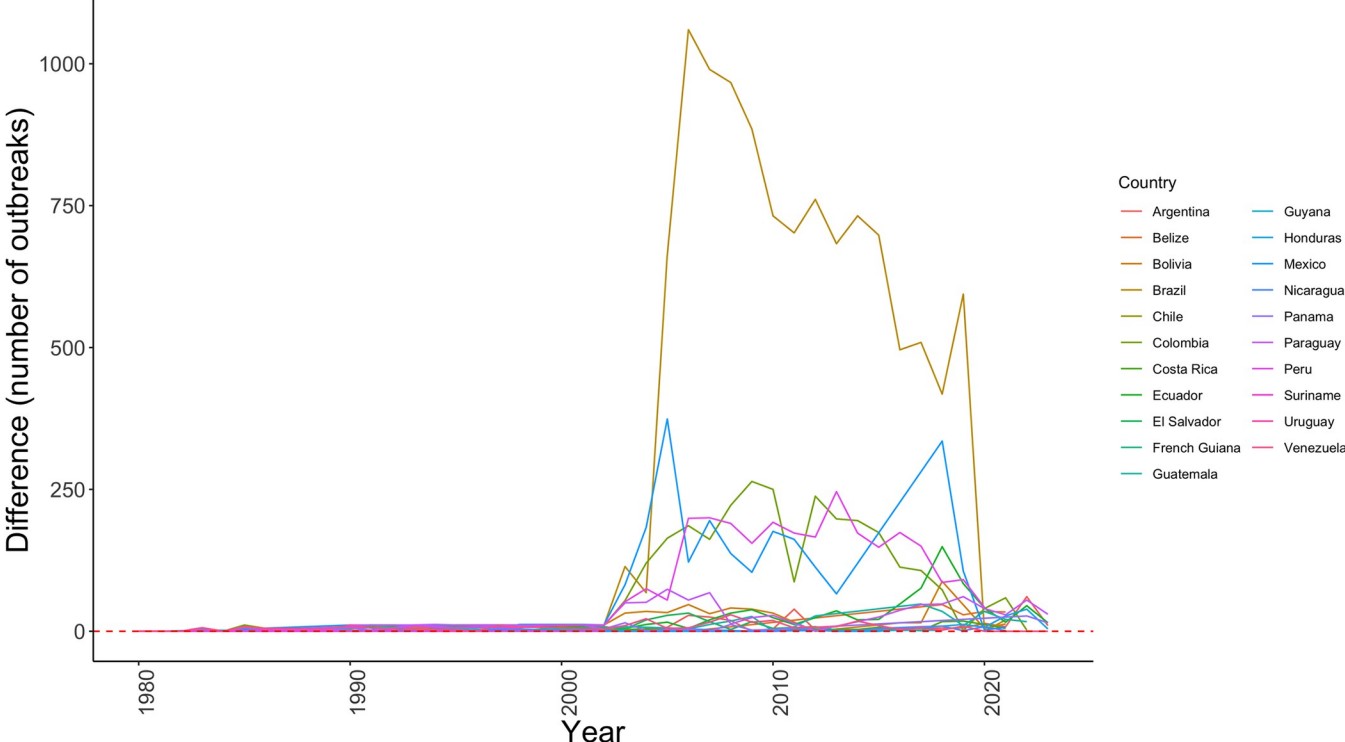

**Fig 3. Change in outbreak incidence across time.** A baseline of outbreaks during the 1970–1979 decade (red dashed line) was used to estimate changes in the annual frequency of rabies outbreaks by country (solid lines). A value equal to 0 on the *y*-axis denoted no change, while values >0 were an increase and <0 were a decrease in the number of annual outbreaks reported per country. Note that all countries consistently showed an increase in outbreaks since 2002 compared with reports in the 1970s.

## Discussion

We evaluated the main epidemiological patterns of rabies incidence in cattle attributed to *D. rotundus* in Latin America between 1970–2023. Understanding trends in *D. rotundus* RABV cross-species transmission from bats to cattle is essential for public health, animal health, wildlife conservation, and the overall well-being of rural communities [39]. For public health, updates on the epidemiology of rabies allows the development of effective interventions, protecting human populations at risk, and mitigating the economic and social impacts of the disease [40]. In the case of animal health, understanding the burden of rabies allows the development of specific interventions in livestock production systems, promotes animal well-being, and supports sustainable agriculture by reducing the economic impact of rabies on livestock herds in Latin America [41, 42]. In wildlife, more information on rabies circulation can help guide epidemiological surveillance in wildlife, can help protect wild populations affected by disease, and can mitigate uninformed bat culling for the control of rabies [43]. Culling non-target bat species for rabies control actually could increase rabies transmission with the collateral damage of decrease of bat-derived ecosystems services [3, 44].

No seasonality was detected in rabies cross-species transmission events. Instead, a constant incidence was observed in the frequency of rabies outbreaks in cattle in Latin America. Rabies outbreaks in cattle have increased at different magnitudes in Latin American countries in the last 20 years. Outbreak size (cases reported per outbreak), however, has decreased. The lack of detectable seasonal fluctuations in outbreak reports suggests the absence of a detectable effect of weather on the continental patterns of rabies cross-species transmission [45]. At the local

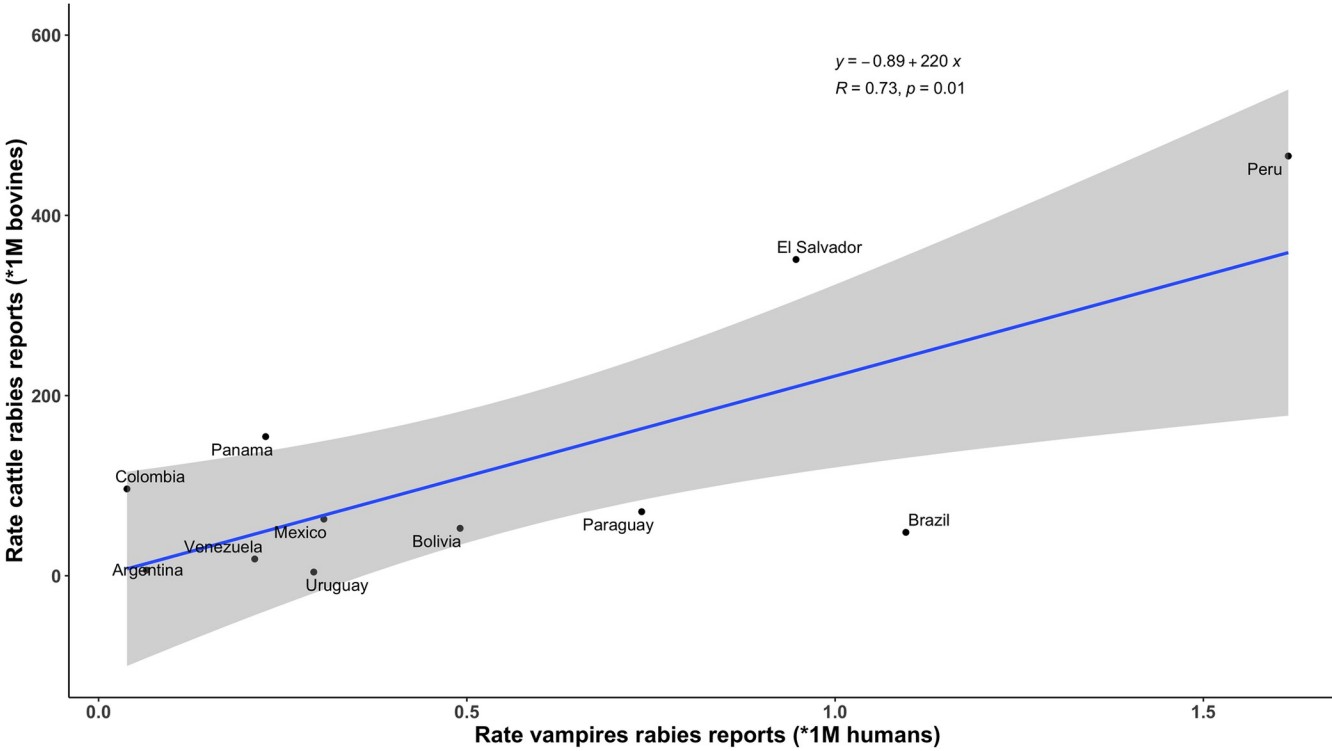

**Fig 4. Relationship between rabies in *Desmodus rotundus* and rabies in cattle.** Correlation between the incidence of rabies in *D. rotundus* (*x*-axis) and rabies outbreaks in cattle (*y*-axis) in Latin America during 1970–2023 ($R = 0.73$, $p = 0.01$). To account for country size, the number of cases of rabies in *D. rotundus* was standardized per million inhabitants and the number of outbreaks of rabies in cattle was standardized per the amount of cattle per country. Blue line: linear model. Black points: countries with detection of rabies virus in *D. rotundus*.

level, however, there may be periods in which there is a remarkable increase in the number of rabies cases likely linked to weather [46, 47].

Rabies outbreaks could be triggered by various factors, such as changes in environmental conditions, interactions between wildlife and livestock, and a decrease in rabies vaccination in cattle [48]. Mandatory vaccination of cattle in high-risk foci and peripheral areas and voluntary vaccination in low-risk areas is suggested as a strategy for rabies control in some countries [39]. In the Americas, mandatory vaccination against rabies for livestock it is not applied uniformly. The dynamics of rabies transmission can be influenced by changes in the demography of bats, livestock management practices, and local habitat [49, 50]. Here, we report a positive association between the number of rabies outbreaks in *D. rotundus* and the number of rabies outbreaks in cattle. This suggests that wildlife health impacts domestic animal health. That is, the wildlife-livestock interface can be modulated by external forces, so that monitoring wildlife health benefits the livestock industry [51].

In other studies, the behavior of *D. rotundus*, such as feeding and movement patterns, have not revealed significant monthly variations [29, 31]. In tropical Latin America, environmental conditions remain relatively stable throughout the year, with minimal variations in temperature, precipitation, or other factors influencing the behavior and demography of bats and livestock [52]. The absence of significant changes in livestock management over the months could be explained by the non-seasonality in rabies transmission patterns in Latin America [39]. Despite the non-significant difference in monthly outbreaks, at the local level, months like May could coincide with favorable conditions for greater activity or reproduction of bats [53].

Thus, the lack of detection of monthly trends could be influenced by biased data collection or surveillance [46, 54], and incomplete data obscuring detectable signals of seasonality. Alternatively, stable food availably offered by livestock herds could help sustain *D. rotundus* abundances and rabies incidence unchanged. Possible seasonal variations in the behavior of *D. rotundus*, including feeding patterns, resting preferences, RABV seroprevalence, and movement and migration patterns, as well as demographic changes, should be contrasted with rabies in cattle in future research.

Future research could also explore the influence of environmental factors, such as temperature, humidity, and vegetation structure, on *D. rotundus* habitat use and their interaction with livestock. Similarly, evaluations of seasonal patterns in livestock movement, behavior, and grazing practices could help inform their vulnerability to *D. rotundus* bites. The SIRVERA dataset used here could also be informative for continent-wide models of rabies transmission dynamics within *D. rotundus* populations and between *D. rotundus* and cattle herds to predict possible seasonal, latitudinal, and altitudinal trends.

Tropical environments are home to diverse ecosystems with a great wealth of wildlife, which can be prey for *D. rotundus* [15, 55]. *Desmodus rotundus* prey can modulate the maintenance and transmission of RABV in an ecosystem via dilution and amplification effects [19]. We found that tropical countries had the highest number of outbreaks across the study period (Figs 1 and 2A). In the intertropical belt of the Americas, *D. rotundus* is distributed in greater abundance and in a greater number of colonies due to favorable climatic conditions [31]. Similarly, dense vegetation and varied habitats in tropical regions provide suitable roosting sites for *D. rotundus* and facilitate their access to livestock [56]. Traditional livestock management practices in tropical regions, such as free-ranging livestock and communal grazing areas, may increase the likelihood of interactions between livestock and *D. rotundus* [50, 57]. We argue that our results suggest that traditional livestock management in the tropics should be revisited.

This study was conducted with data from passive epidemiological surveillance, with has implicit sampling bias. For example, some Latin American countries face challenges in terms of limited veterinary resources, including vaccination programs and surveillance [58]. A lack of resources can make it difficult to effectively detect, quantify, control, and prevent *D. rotundus* rabies in cattle. Limited awareness and education about rabies and its transmission dynamics may contribute to delays in responses and inadequate preventive measures [39, 59], resulting in an underestimation of the burden of *D. rotundus* rabies. Livestock farmers may be less informed about the risks and ways to protect livestock from rabies in the tropics [60]. Our direct observation also revealed that in countries like Guatemala, due to limitations in resources and logistics, field veterinary epidemiologists often send to the laboratory less samples (heads of cattle) than the actual number of animals deceased (*Escobar LE, personal communication 1/10/2024*). Thus, given the differences in diagnostic capabilities across Latin America [9], future research could assess the burden of *D. rotundus* RABV accounting for biases in disease surveillance, laboratory capacities, and reporting systems [57, 61]. We argue that larger outbreaks could be underestimated due to surveillance fatigue, where limited field and laboratory resources limit capacities to track the actual number of deceases animals [62].

Underreporting of cattle rabies cases in Latin America can be a major concern, influenced by factors such as surveillance limitations, lack of awareness, and socioeconomic conditions. Existing rabies surveillance systems often rely on passive reporting by livestock farmers, which can lead to underreporting in remote areas [63]. Local livestock farmers and veterinarians may lack training to identify rabies, resulting in missed cases in aeras where the disease is not endemic [64]. In rural Latin America, economic pressures may discourage reporting by farmers to avoid potential economic losses [39]. Limited access to veterinary services in rural areas

may hamper timely diagnosis and reporting [47]. Low awareness of rabies in some areas contributes to underreporting [65]. We argue that education and training should be conducted in the areas identified in this study. Improved education and training of livestock farmers and veterinary staff are key to improving reporting rates [65].

Additionally, underreporting of bovine rabies cases in Latin America has significant implications for public health, economic stability, and disease management strategies. Underreporting leads to an underestimation of livestock deaths, with studies suggesting that for every reported case, there may be 4.6 unreported cases [16]. Economic losses attributed to bovine rabies can reach up to $171,992 per year in specific regions, significantly affecting smallholder farmers [16]. The persistence of rabies in livestock, particularly from *D. rotundus*, poses a direct risk to human health, as evidenced by the high percentage of human cases linked to rabies in livestock [65]. Lack of consistent reporting hampers the ability to track rabies outbreaks, complicating public health responses [66]. Underreporting of cases leads to poorly informed vaccination strategies, which may not adequately address the true scale of the problem [46, 67]. Improved surveillance and reporting mechanisms are essential to accurately assess the disease burden and implement effective control measures [65].

Future research to address underreporting of bovine rabies cases in Latin America should focus on improving surveillance systems, enhancing farmer education, and using advanced modeling techniques. Transitioning from passive to active surveillance methods to ensure timely reporting of cases, as seen in Ecuador's AGROCALIDAD program [63]. Similarly, implementing GIS tools to map and analyze spatiotemporal trends of rabies transmission [47]. Educating farmers about rabies symptoms and reporting protocols could help increase case identification [16]. Involving local communities in surveillance efforts to improve reporting rates, particularly in remote areas, can help detect cryptic cases of wildlife rabies [16].

Annual outbreaks of rabies in cattle differed significantly between countries. The countries with the highest number of outbreaks per year were Brazil, Colombia, Mexico, and Peru. We found that these countries should consider sustained investments in rabies vaccination, *D. rotundus* RABV research, and rabies surveillance under the One Health approach. This integral rabies monitoring could help secure healthier people and livestock, which directly benefits public health and food security. Countries with strong educational and vaccination programs, coupled with community participation may experience better rabies control [68]. Differences in the ability to implement and maintain national rabies prevention and control measures, such as vaccination [39, 52], may be influencing the geographic variations in the frequency of outbreaks reported here. Future studies should investigate socioeconomic factors that may contribute to differences in local rabies incidence, outbreak size, and reporting within and between countries.

The prevalence of rabies in cattle in Brazil, Colombia, Mexico, and Peru can be attributed to interrelated factors, such as the distribution of *D. rotundus*, cattle density, and environmental conditions. These countries present ecological and socioeconomic characteristics that facilitate sustained rabies transmission among cattle. The density *of D. rotundus* populations correlates with rabies outbreaks [69]. Brazil and Colombia have significant populations of *D. rotundus*, which thrive in rural areas where cattle are abundant [47]. High cattle populations in these countries increase the risk of rabies transmission. For example, Colombia reported 4888 confirmed cases of cattle rabies between 2005 and 2019, with significant outbreaks in regions with high cattle density [47, 52]. Rabies cases in Mexico were concentrated in states with high cattle densities, indicating a direct relationship between cattle density and outbreak frequency [69]. Climatic and topographic conditions in these countries create favorable environments for *D. rotundus*, enhancing its interaction with cattle [69]. Socioeconomic factors influence

inadequate vaccination coverage and limited public awareness, which contribute to the persistence of rabies outbreaks despite control efforts [52, 67].

Data revealed an increase in rabies outbreaks in all Latin American countries starting in 2002. The increase in the number of outbreaks could suggest more frequent cross-species transmission events or improved surveillance capabilities in the region. The geographic expansion of *D. rotundus* and the continent-wide increase in outbreaks reveal more frequent cross-species spillover transmission events of a bat-borne virus. For example, *D. rotundus* populations have been favored by anthropogenic activities, such as the widespread land use changes in Latin America for livestock production [51, 70]. The abundance and accessibility of livestock as a food source for *D. rotundus* increase the subside of resources for the bats and increase their contact with cattle and the possibility of RABV cross-species transmission [15, 51, 71]. The destruction of *D. rotundus* colonies is linked to bats' displacement facilitating rabies spread to novel regions [19, 23, 49, 72]. All countries revealed a decrease in outbreak reports in the early 2020s, which possibly was due to the COVID-19 pandemic limiting surveillance. For example, in Germany, there were fewer reports of upper respiratory tract infections, gastrointestinal infections, and urinary tract infections during the COVID-19 pandemic [73], as there were for Lyme disease in the United States [74]. Particularly for rabies, in India, the COVID-19 lockdown indirectly reduced the reporting of dog bite cases [75], and in the United States, both the number of samples sent for rabies diagnosis and the number of reported rabies cases decreased during 2020 [76] and 2021 [77].

The number of infected animals in each outbreak varied geographically (Table 1 and Fig 1) and temporally (Fig 2). We found a decrease in outbreak size (number of cases per outbreak), which could suggest that epidemic management strategies have been more effective to control outbreaks. The implementation of educational programs on animal health and rabies for livestock farmers in rural areas has been associated with a decrease in the incidence of rabies in cattle [39]. Improved diagnosis allows faster and more accurate identification of RABV, leading to timely rabies control in cattle [59] and humans [78]. More accessible rabies vaccination of livestock, pre and post exposure, leads to a decrease in the number of rabies cases (Ribeiro et al., 2021), while lower vaccination coverage has been linked to a rise in rabies cases [16, 52]. Thus, improving cattle vaccination coverage in both endemic and recently infected areas is a robust tool to reduce outbreak size in livestock [16, 52].

There was a positive association between rabies outbreaks in *D. rotundus* and outbreaks in cattle (Fig 4). The model revealed that Peru, El Salvador, Panama, and Colombia showed more rabies outbreaks in cattle than expected from the number of rabid *D. rotundus* reported maybe due to their high livestock densities. This analysis was geographically limited to the relatively small number of *D. rotundus* samples submitted to national veterinary diagnostic laboratories. In this sense, campaigns for the control and surveillance of rabies in livestock species should include the monitoring of RABV in *D. rotundus* populations [28]. Previous models have shown that livestock abundance is linked to *D. rotundus* abundance and that *D. rotundus* abundance is linked to rabies incidence in *D. rotundus*, which correlated with RABV cross-species transmission to livestock [28, 51, 79]. More than half (52.4%) of Latin American countries reported no cases of rabies in *D. rotundus* (e.g., Guyana, French Guiana, Suriname), suggesting limited to nil RABV surveillance in wildlife reservoirs or poor reporting. The geographic expansion of *D. rotundus* facilitating its invasion to new geographies leads to a greater risk of the emergence of rabies [80, 81]. As such, *D. rotundus* monitoring and surveillance should be at the front of rabies prevention and control. For example, countries such as the United States [76, 77, 82, 83], Mexico [54], Costa Rica [84], Peru [16], and Brazil [85, 86] have bat-borne rabies monitoring programs. The basic components to be considered in a bat-borne rabies monitoring program include

surveillance in rural and urban areas, training on bat-borne rabies for stakeholders and the public, prophylactic measures for exposed people, awareness campaigns in cooperation with environmental agencies in relation to legislation on hunting, breeding and marketing of wild animals, and community education on the risks of zoonotic diseases transmitted by wildlife [85, 86].

*Desmodus rotundus* can pose a serious threat to public health, particularly in rural and indigenous communities where interactions with bats are more frequent. For example, in 2011, an outbreak in Yupicusa, Peru, resulted in the deaths of 21 children and two adults, highlighting the severe impact of *D. rotundus*-transmitted rabies on vulnerable populations [8]. A recent outbreak among the Maxakali people in Brazil resulted in the death of four children due to recreational contact with bats, highlighting cultural practices that increase the risk of exposure [87].

Historically, management of bat-borne rabies involves pre-exposure vaccination of livestock, vaccination of humans bitten by *D. rotundus*, and culling of bats [20, 23]. Bat culling includes the topical use of anticoagulant poisons, such as warfarin or diphacinone [24, 27, 28]. Bat culling, however, has not proven to effectively serve to reduce the burden of rabies. Instead, bat culling is counterproductive as it can increase the circulation of RABV in the area [3, 49]. Furthermore, modification of livestock practices is suggested to reduce the accessibility of bats to their food source [88]. For example, the use of artificial light, protective pens, modification of the size, composition and location of the livestock herd, or hormonal reproductive control of bats has been proposed [20, 27]. The effectiveness of these measures has not been rigorously evaluated. An oral vaccine ingested and transferred from bat to bat can reduce costs and increase coverage at the population scale [3, 20, 89], but their effect on the ecology of rabies and the demographics of *D. rotundus* have not been explored.

One of the goals of the World Health Organization and Pan American Health Organization is that rabies control must be multinational by the year 2030 [90]. In this regard, future rabies research can explore detailed distributional trends of *D. rotundus* range expansion, including its speed, direction, and corridors used for expansion. Understanding the demography and distribution of *D. rotundus* could help delineate cattle herds at risk of rabies cross-species transmission in new regions.

A limitation of this study was the spatial uncertainty derived from the lack of standardization and data gaps at the department, state, and municipality levels in the SIRVERA database. A proportion of rabies in livestock records had administrative data at the state, province, or department, and less at the municipal level. Only recent SIRVERA records offered greater data quality suggesting a relative bias and lack of information in older data plagued by null data. Comparatively, countries such as Brazil, Colombia, Mexico, and Peru had more complete records, while data from countries such as French Guyana and Suriname were generally lacking. Gaps in the dataset highlight the inequities in public and veterinary public health agencies across the Americas, with more developed countries hosting PAHO/PANAFTOSA showing better surveillance systems. The next frontier in rabies prevention and control in Latin America should consider data generation, compilation, and sharing at finer spatial scale than the national level, such as the municipality or farm level. Finer geographic scales would allow more precise georeferenced data and more informative analysis than those feasible with the current database. SIRVERA, however, depends on voluntary reports made by animal health authorities or national veterinary services in Latin American countries. The quality of the data available per country in SIRVERA could serve as a diagnostic tool for the maturity and accuracy of veterinary surveillance systems in the region and could help inform adaptive management and capacity-building efforts.

## Conclusions

Understanding the regional patterns of RABV cross-species transmission from *D. rotundus* to cattle facilitates effective continent-wide control and prevention strategies. We found that rabies is undergoing spread with more outbreaks registered across time and some countries being more severely affected. We found that the widespread distribution of *D. rotundus* RABV and a year-long incidence complicate rabies control effort. Results presented here could help determine where and when monitoring of *D. rotundus* populations and active monitoring of RABV in bats should be prioritized. *Desmodus rotundus* RABV could be a robust model to study bat-borne virus cross-species transmission to understand the drivers of recent and future epidemics of bat-borne viruses. Rabies in livestock is a food security problem that largely affects rural communities and threatens the well-being of farmers, public health, and veterinary health. This study reveals a rise in the number of annual outbreaks of bat-borne rabies in cattle across Latin America.

## Supporting information

**S1 Fig. Temporal trend over decades of rabies in cattle in Latin America. A)** Number of rabies outbreaks in cattle accumulated per decade, showing an increase in outbreaks starting in the 2000s with a drop in the early 2020s. **B)** Number of cases of rabies in cattle accumulated per decade, showing a decrease in cases starting in the 2000s.
(TIFF)

**S2 Fig. Relationship between outbreaks and annual cases of cattle rabies in Latin America.** Correlation between the average annual outbreaks (*x*-axis) and the average annual cases of rabies in cattle (*y*-axis) in Latin American countries during 1970–2023. Blue line: linear model. Black spots: countries with outbreaks and cases of rabies in cattle. Statistics: R = 0.96 and p = 2.2 x $10^{-12}$.
(TIFF)

**S1 Table. Summary by country of reports of rabies in cattle and vampire bats and estimated rates.**
(DOCX)

## Acknowledgments

We thank Marco Antonio Natal Vigilato and Felipe Rocha for facilitating access to SIRVERA data.

## Author Contributions

**Conceptualization:** Luis E. Escobar.

**Data curation:** Diego Soler-Tovar.

**Formal analysis:** Diego Soler-Tovar.

**Funding acquisition:** Luis E. Escobar.

**Investigation:** Diego Soler-Tovar.

**Methodology:** Diego Soler-Tovar, Luis E. Escobar.

**Supervision:** Luis E. Escobar.

**Visualization:** Diego Soler-Tovar.

Writing – **original draft:** Diego Soler-Tovar.

Writing – **review & editing:** Diego Soler-Tovar, Luis E. Escobar.

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
