## [Decision Letter · Decision Letter 0]

25 Apr 2024

PONE-D-24-15035Rabies transmitted from vampires to cattle: An overviewPLOS ONE

Dear Dr. Soler-Tovar,

Thank you for submitting your manuscript to PLOS ONE. After careful consideration, we feel that it has merit but does not fully meet PLOS ONE’s publication criteria as it currently stands. Therefore, we invite you to submit a revised version of the manuscript that addresses the points raised during the review process.

We look forward to receiving your revised manuscript.

Kind regards,

Julian Ruiz-Saenz

Academic Editor

PLOS ONE

Journal Requirements:

“LEE was supported by National Science Foundation CAREER (2235295) and HEGS (2116748) awards. Research reported in this publication was supported by the National Institute of Allergy and Infectious Diseases of the National Institutes of Health under Award Number K01AI168452. The content is solely the responsibility of the authors and does not necessarily represent the official views of the National Institutes of Health.”

Reviewers' comments:

Reviewer's Responses to Questions

**Comments to the Author**

1. Is the manuscript technically sound, and do the data support the conclusions?

Reviewer #1: No

Reviewer #2: Yes

2. Has the statistical analysis been performed appropriately and rigorously? 

Reviewer #1: Yes

Reviewer #2: I Don't Know

3. Have the authors made all data underlying the findings in their manuscript fully available?

Reviewer #1: Yes

Reviewer #2: No

**4. Is the manuscript presented in an intelligible fashion and written in standard English?**

**Reviewer #1: No**

Reviewer #2: Yes

5. Review Comments to the Author

**Reviewer #1: **

I consider that this manuscript provides useful information and quantified data that can be considered to improve the epidemiological surveillance of rabies in Latin America.

Some of my comments and suggestions are

In the introduction, line 118, the authors should rewrite the objective and review whether the appropriate word evaluate could be modified to identify or determine the association between the number of D. rotundus infected with RABV and confirmed cases of rabies in the livestock in Latin America.

They also mention that they are considering the sustained expansion of agriculture in this region and nowhere in the manuscript do they mention how they considered it, I did not find at least figures, or territorial extensions that show cases in areas with this factor. If this was considered in the study, it is necessary to add it to Materials and methods.

line 158, you need to add a bibliographic reference in the text that supports the described methodology and add it to the references section

In line 414, the authors justify the association between rabies outbreaks in D. rotundus and cattle with the density of cattle, however they do not refer to a figure for the number of animals that could strengthen the idea. Furthermore, the authors could consider that this association is most likely due to the small number of samples from D. rotundus sent to the laboratory. In this case, it is important to refer to whether the campaign for the control and surveillance of rabies in livestock species has as its strategy the control of populations in D. rorundus, or whether it would be an area to strengthen in Peru and Colombia.

lines 417 and 418. Review the citations in the text and approve them according to the journal's guidelines.

n the Materials and Methods section, specify whether only cases in cattle or also in other livestock species were considered and put cattle in parentheses so that the study population is better understood.

In the materials and methods part, define what was considered a case and what outbreak. Because otherwise the information will be misinterpreted and considered a duplicate, which makes it difficult to understand the graphs in Suplementary Figure 1.

In accordance with the objective set by the authors, it is recommended to add a descriptive table similar to table 1 but with data on the cases that occurred in D.rorundus or those detected with RABV infection, as well as adding a footer to the table in both cases that defines the source of the information and the Diagnostic test they used to confirm them as positive for the disease.

It is necessary to add the conclusions section, it seems that lines 454 to 463 refer to them, however, here it is also necessary to restructure the paragraph to explain how this analysis can be useful to determine where and when should prioritize the monitoring of D. rotundus due to having populations infected with RABV and the probable epidemiological trends of rabies in the near future in Latin America.

In the results section, although the authors mention Figure 1, they do not describe the interpretation of the distribution. Here a paragraph is necessary that states what is observed or makes a comparison between them. It is highly recommended to add a map with the name of the countries to better understand which geographical areas it refers

**Reviewer #2:** 

The manuscript entitled "Rabies transmitted from vampires to cattle: An overview" aimed to describe the occurrence of rabies in Latin America (21 countries) over an extended period (over 50 years) using data obtained from the Pan American Health Organization. The research is highly significant for utilizing a large and hard-to-access dataset, capable of expanding knowledge on the epidemiology of rabies in the rural cycle, where vampire bats are involved.

The introduction is well-written, providing the necessary background for understanding the study, and requires only minor adjustments. The methodology and conducted analyses are also presented, and the results section is extensive, providing interesting data. The conclusion, at the end of the discussion, is supported by the research data.

I suggest some adjustments regarding the manuscript, which are presented in the text below.

Remarks:

Line 50 - Why do the authors use the term "spillover"? It might be better to consider transmission throughout the entire manuscript.

Lines 52-54 – The value of the association should be included, as well as the incidence.

Line 60 – Are there epidemiological analyses to support the mentioned risk factors? The values of the analyses should be included. If not, it is speculative and should be either removed or only discussed.

Line 81 – It's not clear what the impacts on the environment would be. The authors need to clarify.

Lines 85-86 – Transmission methods need to be clarified in addition to the virus elimination route.

Line 96 – There is also an important role of wild animals as virus reservoirs.

Line 104 – In Brazil, there has also been a recent increase in the detection of bat variants in humans.

Line 109 – Standardize the use of "D. rotundus" (abbreviated) after the first mention in the introduction.

In the introduction, I suggest the authors include data on the economic impacts of rabies resulting from its occurrence in production animals (cattle).

Line 140 – I would like to understand how notifications are entered into SIRVERA. Who provides this regional data? Who receives the notifications and forwards them to the Pan American Health Organization? The logistics of obtaining and including data need to be clear to readers.

I would also like to understand in the methodology how the authors calculated the incidence values mentioned in the abstract. Was there a collection of the population size of the herds in the evaluated areas? Or were only the case numbers considered without taking into account the size of the cattle herds in the areas to consider true incidence?

In Figure 1A, I see countries inconsistent with the scale presented. Only two colorations? What about the others? Why only include the extremes (larger and smaller)? I would like to visualize the data in its entirety through the map.

Also, the image definition is quite poor for visualization.

Lines 190-194 – It should not be considered as seasonal variation since it only considers the time of year (month) without taking into account the other characteristics that determine seasonality.

Line 228 – In which species?

Line 297 – It would be important to discuss the mandatory vaccination of cattle in different countries.

Line 351 – Reference 30 also shows the importance of sanitary education actions, as well as population control of hematophagous bats and vaccination as tools for rabies control in herbivores.

Lines 372-374 – There are works in Brazil that have evaluated the cost-benefit effect of applying vaccines for rabies control in herbivores (https://doi.org/10.1590/1678-5150-PVB-6201).

In the discussion, I would also like to see information on population control measures of hematophagous bats, such as the application of vampiricidal pastes. It is known that these actions have good efficacy but can also pose a high risk to non-hematophagous bat species. A discussion on these actions is important to include.

Lines 405; 417-418 and others – Adjust the reference to the number.

Lines 423-424 - The existence of rabies monitoring programs in bats should be discussed. Which countries do this? Is there a requirement for this monitoring? The topic can be better detailed.

6. PLOS authors have the option to publish the peer review history of their article (what does this mean?). If published, this will include your full peer review and any attached files.

Reviewer #1: No

Reviewer #2: No

---

## [Author Response · Author response to Decision Letter 0]

23 Jun 2024

Response to Reviewers

PONE-D-24-15035

Below is shown each comment from the Academic Editor and the reviewers, and after each one the response from the authors.

Academic Editor Comments

Authors' response: The suggested links have been re-checked and the manuscript meets the style requirements of PLOS ONE.

2. Thank you for stating in your Funding Statement: “LEE was supported by National Science Foundation CAREER (2235295) and HEGS (2116748) awards. Research reported in this publication was supported by the National Institute of Allergy and Infectious Diseases of the National Institutes of Health under Award Number K01AI168452. The content is solely the responsibility of the authors and does not necessarily represent the official views of the National Institutes of Health.” Please provide an amended statement that declares *all* the funding or sources of support (whether external or internal to your organization) received during this study, as detailed online in our guide for authors at http://journals.plos.org/plosone/s/submit-now. Please also include the statement “There was no additional external funding received for this study.” in your updated Funding Statement. Please include your amended Funding Statement within your cover letter. We will change the online submission form on your behalf.

Authors' response: The manuscript and data submitted on the website were reviewed, and all funds or sources of support received during the study were included; additionally, “No additional external funding was received for this study” was included in the statement. The modification of the financing statement was included in the cover letter, as follows: LEE was supported by National Science Foundation CAREER (2235295) and HEGS (2116748) awards. Research reported in this publication was supported by the National Institute of Allergy and Infectious Diseases of the National Institutes of Health under Award Number K01AI168452. The content is solely the responsibility of the authors and does not necessarily represent the official views of the National Institutes of Health. No additional external funding was received for this study.

3. We note that Figure 1 in your submission contain [map/satellite] images which may be copyrighted. All PLOS content is published under the Creative Commons Attribution License (CC BY 4.0), which means that the manuscript, images, and Supporting Information files will be freely available online, and any third party is permitted to access, download, copy, distribute, and use these materials in any way, even commercially, with proper attribution. For these reasons, we cannot publish previously copyrighted maps or satellite images created using proprietary data, such as Google software (Google Maps, Street View, and Earth). For more information, see our copyright guidelines: http://journals.plos.org/plosone/s/licenses-and-copyright. We require you to either (1) present written permission from the copyright holder to publish these figures specifically under the CC BY 4.0 license, or (2) remove the figures from your submission:

Authors' response: The map in Figure 1 is not previously copyrighted and is not a satellite image created with data from Google software (Google Maps, Street View, and Earth), it is a map created by the authors in R and RStudio, using shapefiles from the sources suggested by PLOS ONE journal.

Reviewers' Comments

1. Is the manuscript technically sound, and do the data support the conclusions?

Reviewer #1: No

Reviewer #2: Yes

Authors' response: We reviewed and adjusted the manuscript according to the comments of the Academic Editor and the reviewers and considered that the manuscript describes technically sound scientific research with data supporting the conclusions, the analyses were done rigorously, and the conclusions were generated from the results presented.

2. Has the statistical analysis been performed appropriately and rigorously?

Reviewer #1: Yes

Reviewer #2: I Don't Know

Authors' response: We confirm that the statistical analysis was carried out appropriately and rigorously

3. Have the authors made all data underlying the findings in their manuscript fully available?

Reviewer #1: Yes

Reviewer #2: No

Authors' response: The data used in the analysis for this manuscript are completely available without restrictions, on the SIRVERA website: https://sirvera.panaftosa.org.br/. In addition, the R code used for the analyses will be deposited in a public repository (like GitHub: https://github.com/) and will also be delivered by email without restrictions by the authors to anyone who requests them.

4. Is the manuscript presented in an intelligible fashion and written in standard English?

Reviewer #1: No

Reviewer #2: Yes

Authors' response: Comments on English from the reviewers were considered, the English of the manuscript was reviewed by members of the Virginia Tech laboratory (United States) and the Grammarly tool was used to improve the English.

5. Review Comments to the Author

Reviewer #1: 

1. In the introduction, line 118, the authors should rewrite the objective and review whether the appropriate word evaluate could be modified to identify or determine the association between the number of D. rotundus infected with RABV and confirmed cases of rabies in the livestock in Latin America.

Authors' response: The objective of the manuscript was modified according to the reviewer's comment.

2. They also mention that they are considering the sustained expansion of agriculture in this region and nowhere in the manuscript do they mention how they considered it, I did not find at least figures, or territorial extensions that show cases in areas with this factor. If this was considered in the study, it is necessary to add it to Materials and methods.

Authors' response: The sustained expansion of agriculture in the region was not considered either methodologically or analytically, but this text is included followed by the objective to justify it and is supported with a reference.

3. Line 158, you need to add a bibliographic reference in the text that supports the described methodology and add it to the references section.

Authors' response: References 30 and 31 were included.

4. In line 414, the authors justify the association between rabies outbreaks in D. rotundus and cattle with the density of cattle, however they do not refer to a figure for the number of animals that could strengthen the idea. Furthermore, the authors could consider that this association is most likely due to the small number of samples from D. rotundus sent to the laboratory. In this case, it is important to refer to whether the campaign for the control and surveillance of rabies in livestock species has as its strategy the control of populations in D. rorundus, or whether it would be an area to strengthen in Peru and Colombia.

Authors' response: S1 Table with the quantitative data that supports Figure 4 was included as Supporting information. In addition, the analysis was run again to adjust the statistical values and include El Salvador in the text. In addition, text was included in the Discussion, as suggested, supported by reference 21.

5. Lines 417 and 418. Review the citations in the text and approve them according to the journal's guidelines.

Authors' response: The citations were reviewed and comply with the journal's guidelines.

6. In the Materials and Methods section, specify whether only cases in cattle or also in other livestock species were considered and put cattle in parentheses so that the study population is better understood.

Authors' response: In lines 136 and 141 it is indicated that the data are for rabies in cattle.

7. In the materials and methods part, define what was considered a case and what outbreak. Because otherwise the information will be misinterpreted and considered a duplicate, which makes it difficult to understand the graphs in Suplementary Figure 1.

Authors' response: In lines 148 and 149 it is indicated that it is a case and that it is an outbreak.

8. In accordance with the objective set by the authors, it is recommended to add a descriptive table similar to table 1 but with data on the cases that occurred in D. rotundus or those detected with RABV infection, as well as adding a footer to the table in both cases that defines the source of the information and the Diagnostic test they used to confirm them as positive for the disease.

Authors' response: Some of the additional data suggested by the reviewer was included in Table S1 of the Supporting information. The source of the data in Table 1 is included at the end of that table's label, and the source of the data in the S1 Table is included at the end of the table with asterisks.

9. It is necessary to add the conclusions section, it seems that lines 454 to 463 refer to them, however, here it is also necessary to restructure the paragraph to explain how this analysis can be useful to determine where and when should prioritize the monitoring of D. rotundus due to having populations infected with RABV and the probable epidemiological trends of rabies in the near future in Latin America.

Authors' response: The Conclusions subtitle was included, and that section was enriched according to the reviewer's comment.

10. In the results section, although the authors mention Figure 1, they do not describe the interpretation of the distribution. Here a paragraph is necessary that states what is observed or makes a comparison between them. It is highly recommended to add a map with the name of the countries to better understand which geographical areas it refers.

Authors' response: The most relevant part of Figure 1 is included in lines: 165-173, 184-188, and 405-406. The names of the countries were included in the maps in Figure 1.

Reviewer #2: 

1. Line 50 - Why do the authors use the term "spillover"? It might be better to consider transmission throughout the entire manuscript.

Authors' response: The word "spillover" was changed to "cross-species transmission" throughout the manuscript.

2. Lines 52-54 – The value of the association should be included, as well as the incidence.

Authors' response: The suggested values were included in the indicated lines.

3. Line 60 – Are there epidemiological analyses to support the mentioned risk factors? The values of the analyses should be included. If not, it is speculative and should be either removed or only discussed.

Authors' response: Some of these factors are discussed in the Discussion.

4. Line 81 – It's not clear what the impacts on the environment would be. The authors need to clarify.

Authors' response: This comment was answered on line 88, with their reference.

5. Lines 85-86 – Transmission methods need to be clarified in addition to the virus elimination route.

Authors' response: Lines 91 and 92 were included with their reference.

6. Line 96 – There is also an important role of wild animals as virus reservoirs.

Authors' response: Text was included on lines 102 and 103, along with references.

7. Line 104 – In Brazil, there has also been a recent increase in the detection of bat variants in humans.

Authors' response: In addition, the detection of Text and references were included in lines 113 and 114 to respond to this comment. has recently increased.

8. Line 109 – Standardize the use of "D. rotundus" (abbreviated) after the first mention in the introduction.

Authors' response: The use of “D. rotundus” (abbreviated) after the first mention in the introduction was revised and standardized, only leaving the full name if it starts a paragraph or if named after a point in a paragraph.

9. In the introduction, I suggest the authors include data on the economic impacts of rabies resulting from its occurrence in production animals (cattle).

Authors' response: A paragraph about the economic impact was included between lines 120 and 129.

10. Line 140 – I would like to understand how notifications are entered into SIRVERA. Who provides this regional data? Who receives the notifications and forwards them to the Pan American Health Organization? The logistics of obtaining and including data need to be clear to readers.

Authors' response: In lines 151 to 153 what was commented by the reviewer was expanded.

11. I would also like to understand in the methodology how the authors calculated the incidence values mentioned in the abstract. Was there a collection of the population size of the herds in the evaluated areas? Or were only the case numbers considered without taking into account the size of the cattle herds in the areas to consider true incidence?

Authors' response: This comment was clarified in lines 173 to 177.

12. In Figure 1A, I see countries inconsistent with the scale presented. Only two colorations? What about the others? Why only include the extremes (larger and smaller)? I would like to visualize the data in its entirety through the map. Also, the image definition is quite poor for visualization.

Authors' response: New maps were made for Figure 1, improving the color palette, and the resolution or quality of the figure was improved.

13. Lines 190-194 – It should not be considered as seasonal variation since it only considers the time of year (month) without taking into account the other characteristics that determine seasonality.

Authors' response: Changed the word "seasona

---

## [Decision Letter · Decision Letter 1]

14 Aug 2024

PONE-D-24-15035R1Rabies transmitted from vampires to cattle: An overviewPLOS ONE

Dear Dr. Soler-Tovar,

Thank you for submitting your manuscript to PLOS ONE. After careful consideration, we feel that it has merit but does not fully meet PLOS ONE’s publication criteria as it currently stands. Therefore, we invite you to submit a revised version of the manuscript that addresses the points raised during the review process.

We look forward to receiving your revised manuscript.

Kind regards,

Julian Ruiz-Saenz

Academic Editor

PLOS ONE

Reviewers' comments:

Reviewer's Responses to Questions

**Comments to the Author**

1. If the authors have adequately addressed your comments raised in a previous round of review and you feel that this manuscript is now acceptable for publication, you may indicate that here to bypass the “Comments to the Author” section, enter your conflict of interest statement in the “Confidential to Editor” section, and submit your "Accept" recommendation.

Reviewer #2: All comments have been addressed

Reviewer #3: All comments have been addressed

2. Is the manuscript technically sound, and do the data support the conclusions?

Reviewer #2: Yes

Reviewer #3: No

3. Has the statistical analysis been performed appropriately and rigorously? 

Reviewer #2: I Don't Know

Reviewer #3: No

4. Have the authors made all data underlying the findings in their manuscript fully available?

Reviewer #2: Yes

Reviewer #3: Yes

5. Is the manuscript presented in an intelligible fashion and written in standard English?

Reviewer #2: Yes

Reviewer #3: Yes

6. Review Comments to the Author

Reviewer #2: All my suggestions were properly adressed. The manuscript was improve and it is ready for acceptance in my point of view.

**Reviewer #3: **Rabies is an importante zoonosis that affect all mammalians. The work describes data about rabies in cattle transmitted by hematophagous bats Desmodus rotundus. This epidemiological cycle of rabies is very important and causes to much economic losses but the work do not presents new topics about this on the other hand all data is collected in a unique system (SIRVERA). Many points are not discussed as subnotification of cases, acidental cases of rabies in humans transmited by D. rotundus and data from each studied country.

7. PLOS authors have the option to publish the peer review history of their article (what does this mean?). If published, this will include your full peer review and any attached files.

Reviewer #2: No

Reviewer #3: No

---

## [Author Response · Author response to Decision Letter 1]

13 Dec 2024

Response to Reviewers

PONE-D-24-15035R1

Reviewer #3 Comments

Many points are not discussed as: 1) subnotification of cases, 2) accidental cases of rabies in humans transmitted by D. rotundus and 3) data from each studied country.

Authors' response:

Many thanks to reviewer #3 for his valuable comments, a review of the entire manuscript was made, and adjustments were made throughout the text (see attached manuscript).

The three main comments are answered below:

1) We agree with the reviewer regarding the subnotification of cases. We apologize for missing this key factor regarding the patterns presented in the article. We have incorporated new text addressing the effect of the subnotification of cases in the results and overall conclusions. We also updated the discussion and present a new rationale about the likely effects of the underreporting of rabies cases in Latin America. Finally, we present potential alternatives to and future research to address the subnotification in future studies.

The new text is included below (lines 527 to 570 of the manuscript):

“Underreporting of cattle rabies cases in Latin America can be a major concern, influenced by factors such as surveillance limitations, lack of awareness, and socioeconomic conditions. Existing rabies surveillance systems often rely on passive reporting by livestock farmers, which can lead to underreporting in remote areas [63]. Local livestock farmers and veterinarians may lack training to identify rabies, resulting in missed cases in aeras where the disease is not endemic [64]. In rural Latin America, economic pressures may discourage reporting by farmers to avoid potential economic losses [39]. Limited access to veterinary services in rural areas may hamper timely diagnosis and reporting [47]. Low awareness of rabies in some areas contributes to underreporting [65]. We argue that education and training should be conducted in the areas identified in this study. Improved education and training of livestock farmers and veterinary staff are key to improving reporting rates [65].

Additionally, underreporting of bovine rabies cases in Latin America has significant implications for public health, economic stability, and disease management strategies. Underreporting leads to an underestimation of livestock deaths, with studies suggesting that for every reported case, there may be 4.6 unreported cases [16]. Economic losses attributed to bovine rabies can reach up to $171,992 per year in specific regions, significantly affecting smallholder farmers [16]. The persistence of rabies in livestock, particularly from D. rotundus, poses a direct risk to human health, as evidenced by the high percentage of human cases linked to rabies in livestock [65]. Lack of consistent reporting hampers the ability to track rabies outbreaks, complicating public health responses [66]. Underreporting of cases leads to poorly informed vaccination strategies, which may not adequately address the true scale of the problem [46,67]. Improved surveillance and reporting mechanisms are essential to accurately assess the disease burden and implement effective control measures [65].

Future research to address underreporting of bovine rabies cases in Latin America should focus on improving surveillance systems, enhancing farmer education, and using advanced modeling techniques. Transitioning from passive to active surveillance methods to ensure timely reporting of cases, as seen in Ecuador's AGROCALIDAD program [63]. Similarly, implementing GIS tools to map and analyze spatiotemporal trends of rabies transmission [47]. Educating farmers about rabies symptoms and reporting protocols could help increase case identification [16]. Involving local communities in surveillance efforts to improve reporting rates, particularly in remote areas, can help detect cryptic cases of wildlife rabies [16].”

2) We agree with the reviewer in our intentional overlook of accidental cases of rabies in humans transmitted by D. rotundus. Accidental cases of rabies in humans transmitted by D. rotundus were not discussed because our intention with this manuscript was to present the status quo on passive epidemiology about bovine rabies. We intentionally left human cases out because this is under study as part of another PhD thesis chapter at the Escobar lab. We, however, agree with the reviewer in that this is a very interesting topic that deserves attention. To address this suggestion, we have cited literature regarding accidental human cases and have expanded our discussion accordingly.

The new text is included below (lines 711 to 718 of the manuscript):

“Desmodus rotundus can pose a serious threat to public health, particularly in rural and indigenous communities where interactions with bats are more frequent. For example, in 2011, an outbreak in Yupicusa, Peru, resulted in the deaths of 21 children and two adults, highlighting the severe impact of D. rotundus-transmitted rabies on vulnerable populations [8]. A recent outbreak among the Maxakali people in Brazil resulted in the death of four children due to recreational contact with bats, highlighting cultural practices that increase the risk of exposure [87].”

3) We apologize for our limited elaboration on the data from each studied country in the previous version of the manuscript. We have revised the paper following the reviewer’s observation. A new discussion text was included to discuss in more detail incidence and prevalence values at the country level. In addition, rabies reports in cattle and vampire bats and estimated rates are summarized by country in Table S1 of the supporting information.

The new text is included below (lines 587 to 618 of the manuscript):

“The prevalence of rabies in cattle in Brazil, Colombia, Mexico, and Peru can be attributed to interrelated factors, such as the distribution of D. rotundus, cattle density, and environmental conditions. These countries present ecological and socioeconomic characteristics that facilitate sustained rabies transmission among cattle. The density of D. rotundus populations correlates with rabies outbreaks [69]. Brazil and Colombia have significant populations of D. rotundus, which thrive in rural areas where cattle are abundant [47]. High cattle populations in these countries increase the risk of rabies transmission. For example, Colombia reported 4888 confirmed cases of cattle rabies between 2005 and 2019, with significant outbreaks in regions with high cattle density [47,52]. Rabies cases in Mexico were concentrated in states with high cattle densities, indicating a direct relationship between cattle density and outbreak frequency [69]. Climatic and topographic conditions in these countries create favorable environments for D. rotundus, enhancing its interaction with cattle [69]. Socioeconomic factors influence inadequate vaccination coverage and limited public awareness, which contribute to the persistence of rabies outbreaks despite control efforts [52,67].”

---

## [Editor Report · Decision Letter 2]

23 Dec 2024

Rabies transmitted from vampires to cattle: An overview

PONE-D-24-15035R2

Dear Dr. Soler-Tovar,

We’re pleased to inform you that your manuscript has been judged scientifically suitable for publication and will be formally accepted for publication once it meets all outstanding technical requirements.

Kind regards,

Julian Ruiz-Saenz

Academic Editor

PLOS ONE
---

## [Editor Report · Acceptance letter]

2 Jan 2025

PONE-D-24-15035R2 

PLOS ONE

Dear Dr. Soler-Tovar, 

I'm pleased to inform you that your manuscript has been deemed suitable for publication in PLOS ONE. Congratulations! Your manuscript is now being handed over to our production team.

Kind regards, 

on behalf of

Dr. Julian Ruiz-Saenz 

Academic Editor

PLOS ONE